# Viriditoxin Stabilizes Microtubule Polymers in SK-OV-3 Cells and Exhibits Antimitotic and Antimetastatic Potential

**DOI:** 10.3390/md18090445

**Published:** 2020-08-27

**Authors:** Mingzhi Su, Changhao Zhao, Dandan Li, Jiafu Cao, Zhiran Ju, Eun La Kim, Young-Suk Jung, Jee H. Jung

**Affiliations:** 1College of Pharmacy, Pusan National University, Busan 46241, Korea; smz0310@163.com (M.S.); changhao112358@163.com (C.Z.); 18840659614@163.com (D.L.); caojiafu1985@163.com (J.C.); 13719405761@163.com (Z.J.); eunlakim@gmail.com (E.L.K.); youngjung@pusan.ac.kr (Y.-S.J.); 2State Key Laboratory for Functions and Applications of Medicinal Plants, Guizhou Medical University, Guiyang 550014, China

**Keywords:** viriditoxin, tubulin polymerization, microtubule, antimitotic, antimetastatic, SK-OV-3

## Abstract

Microtubules play a crucial role in mitosis and are attractive targets for cancer therapy. Recently, we isolated viriditoxin, a cytotoxic and antibacterial compound, from a marine fungus *Paecilomyces variotii*. Viriditoxin has been reported to inhibit the polymerization of bacterial FtsZ, a tubulin-like GTPase that plays an essential role in bacterial cell division. Given the close structural homology between FtsZ and tubulin, we investigated the potential antimitotic effects of viriditoxin on human cancer cells. Viriditoxin, like paclitaxel, enhanced tubulin polymerization and stabilized microtubule polymers, thereby perturbing mitosis in the SK-OV-3 cell line. However, the morphology of the stabilized microtubules was different from that induced by paclitaxel, indicating subtle differences in the mode of action of these compounds. Microtubule dynamics are also essential in cell movement, and viriditoxin repressed migration and colony formation ability of SK-OV-3 cells. Based on these results, we propose that viriditoxin interrupts microtubule dynamics, thus leading to antimitotic and antimetastatic activities.

## 1. Introduction

Although several anticancer drugs are clinically available, drug therapies for cancer patients are still unsatisfactory owing to the associated undesirable side effects, emergence of drug resistance, and metastasis of cancer cells. Therefore, the development of alternative chemotherapeutic agents with enhanced therapeutic profile is required. Microtubules play a crucial role in mitosis and are recognized as one of the most attractive targets for anticancer drug development [1]. Cancer cells are sensitive to mitotic arrest and often undergo cell death in response to agents that perturb microtubule dynamics.

The tubulin superfamily consists of at least 10 different globular proteins and others may be awaiting discovery [2,3]. Among these, α- and β-tubulins form a heterodimer that polymerizes into microtubules. Microtubules are the major cytoskeletal components of eukaryotic cells and are involved in multiple cellular processes. During interphase, tubulins are polymerized into cytoskeletal microtubules to maintain cell shape, polarity, motility, and intracellular transport. However, during metaphase and anaphase, microtubules are disintegrated into tubulin monomers to be reassembled into microtubules to form mitotic spindles, which are required for chromosome segregation and cell division [4]. 

Agents that target tubulins inhibit microtubule dynamics and interfere with spindle formation, subsequently inhibiting cell proliferation. They are typically represented by plant metabolites, such as colchicine, vinca alkaloids, and taxanes, which induce sustained mitotic arrest at the metaphase/anaphase transition, subsequently causing apoptotic cell death. Recent tubulin-targeting compounds isolated from marine sources include dolastatin 10, dicitrinone D, discodermolide, laulimalide, and peloruside A. These compounds exert selective effects that are correlated with their molecular interactions with tubulins [1,5,6,7]. While searching for cytotoxic compounds against cancer cells from marine sources, we isolated viriditoxin, a cytotoxic compound (Figure 1A), from the jellyfish-derived fungus *Paecilomyces variotii* [8]. Viriditoxin exhibits remarkable cytotoxicity against a number of human cancer cell lines (Table 1) [9,10]. However, the cytotoxic mechanism of viriditoxin has not yet been completely elucidated.

Viriditoxin was reported to inhibit the polymerization of bacterial FtsZ [11]. FtsZ is a tubulin-like GTPase present in prokaryotic cells [12,13]. FtsZ forms polymers and assembles into a cytoskeletal ring at the future site of bacterial cell division in the presence of GTP, and along with additional proteins, the ring constricts to divide the cell in to two daughter cell [14,15,16,17]. The N-terminal GTP-binding domains of tubulin and FtsZ are virtually identical in structure, as expected from the substantial sequence identity. Sequence identity is absent from the C-terminal domains, but they also have virtually identical structures [13]. Given the structural homology of the GTP-binding domains of FtsZ and tubulin, viriditoxin was expected to exhibit antimitotic effect via molecular interactions with the tubulin protein. We therefore attempted to elucidate its cytotoxic and antimitotic mechanisms in the human cancer cell line, SK-OV-3.

## 2. Results and Discussion

### 2.1. Viriditoxin Inhibited Proliferation of SK-OV-3 Cells

For this study, we selected the SK-OV-3 cell line, an ovarian cancer cell line, partly because ovarian cancer is one of the high-risk cancers in women worldwide. The other reason for its selection was that SK-OV-3 is a drug-resistant cancer cell line. According to ATCC, it is resistant to cytotoxic agents, such as diphtheria toxin and tumor necrosis factor, and clinical anti-ovarian cancer drugs, such as cis-platinum and adriamycin [18].

To determine the cytotoxic effects of viriditoxin, SK-OV-3 cells were exposed to viriditoxin or paclitaxel for 24 h. As shown in Figure 1B, viriditoxin significantly inhibited the cell viability in a dose-dependent manner after 24 h of treatment. The IC_50_ value of viriditoxin against SK-OV-3 cells was 14.3 μM, whereas that of paclitaxel was 0.24 μM (Table 1). Moreover, viriditoxin induced marked morphologic changes in the cells after 24 h of treatment. SK-OV-3 cells manifested cytoplasmic shrinkage and cellular flattening. Furthermore, the periphery of cells was rugged with cell fragments when compared with the control group (Figure 1C).

### 2.2. Viriditoxin Enhanced Tubulin Assembly

Microtubules play an important role in cell replication. In the interphase, microtubules act as an internal scaffold that supports the cell shape and structure. During metaphase and anaphase, microtubules are disintegrated back to tubulin monomers only to be reassembled into microtubules to form mitotic spindles, which are required for chromosome segregation and cell division [4]. To examine whether viriditoxin could affect tubulin polymerization in vitro, we compared the behavior of tubulin when exposed to viriditoxin and two antimitotic agents paclitaxel and colchicine. As shown in Figure 2A, viriditoxin (50, and 100 μM) and paclitaxel (10 μM) enhanced tubulin polymerization, whereas colchicine (10 μM) inhibited it. The tubulin polymerization assay is based on the fact that microtubules scatter light to an extent that is proportional to the concentration of microtubule polymers.

As viriditoxin promotes tubulin polymerization in vitro, we next investigated its ability to polymerize cellular tubulin in SK-OV-3 cells. The soluble tubulin and insoluble polymerized tubulin in cells were separated by centrifugation after cell lysis, followed by Western blot analysis. As shown in Figure 2B, approximately equal protein levels of tubulin were observed in control groups. Viriditoxin-treated groups resulted in a shift of soluble tubulin to particulate fractions in a concentration-dependent manner, a finding similar to that observed in the paclitaxel group. In contrast, treatment with colchicine decreased the levels of polymerized tubulin. Collectively, these results suggest that viriditoxin, though it is not as potent as paclitaxel, acts as a microtubule stabilizer both in vitro and in the cell level.

### 2.3. Viriditoxin Stabilized Microtubule Polymers in SK-OV-3 Cells

To further investigate the phenotypic changes in the cytoskeleton network of tubulin, SK-OV-3 cells were incubated with viriditoxin for 4 or 24 h, and the microtubule network of SK-OV-3 cells was visualized using immunofluorescence. As shown in Figure 3, microtubules (green fluorescence) in the cells of the control group were arranged in an orderly and uniform manner. They filled the cytoplasm with an even distribution. We observed time-dependent potentiation of tubulin polymerization after treatment with viriditoxin. Increase in the density of cellular microtubules and formation of long, thick, and disarrayed microtubule bundles surrounding the nucleus indicated enhanced microtubule assembly. This morphological observation was similar to that obtained after treatment with microtubule stabilizer paclitaxel (10 µM, 1 h). The microtubules in these two groups were more extensively polymerized over time to surround the nucleus. On the contrary, colchicine, an inhibitor of microtubule polymerization, decreased the density of cellular microtubules. Moreover, viriditoxin enhanced the tubulin polymerization at a lower concentration of 15 µM after 24 h treatment. Paclitaxel at 10 µM induced massive assembly of microtubule bundles, whereas colchicine at 10 µM induced notable decrease in microtubule filaments (Figure 4). During mitosis, the DNA in the nucleus of the control group replicated into two daughter sets of chromosomes. Microtubules began to form mitotic spindles toward the chromosomes (Figure 5). Paclitaxel generated multipolar spindles preventing equal segregation of DNA in mitotic cells, leading to the arrest of cell division. In contrast, colchicine interrupted the microtubule assembly. Meanwhile, viriditoxin induced abnormal tubulin polymerization with highly dense microtubule linings close to the cell membrane. These pericytoplasmic microtubule structures were apparently distinct from multipolar spindles observed in paclitaxel-treated cells. Viriditoxin-treated cells failed to form the mitotic spindle required for the segregation of chromosomes. Hence, it was assumed that viriditoxin, like paclitaxel, enhanced tubulin polymerization and stabilized the assembled microtubules, thereby suppressing mitosis in SK-OV-3 cells. However, the morphology of the stabilized microtubules was distinct from that of paclitaxel. This could be due to different binding affinities of viriditoxin and paclitaxel to different isotypes of tubulin that would be present in the experimental system. In other words, if we imagine that different isotypes with distinct functions are co-present and one isotype of tubulin forms a mitotic spindle while the other isotype forms microtubules near the cell membrane, and if paclitaxel binds best to the former and viriditoxin to the latter, we may observe the aforementioned morphological difference—or, perhaps, each drug has different effects on the binding of microtubule-associated proteins to the microtubule. Additional biochemical studies should be performed to elucidate the detailed mechanism of viriditoxin on tubulin polymerization and the subtle differences from that of paclitaxel.

### 2.4. Viriditoxin Induced G2/M Phase Arrest and Apoptosis in SK-OV-3 Cells

To examine the effect of viriditoxin on cell cycle progression, cells were treated with several concentrations of viriditoxin (5, 10, 20, and 40 μM), paclitaxel (5 μM), or colchicine (5 μM) for 24 h, followed by flow cytometric analysis. Viriditoxin increased the population of cells in the G_2_/M phase and concomitantly decreased the population of cells in the G_1_ phase in SK-OV-3 cells in a concentration-dependent manner (Table 2). The proportion of S phase of the cells treated with viriditoxin was increased in proportion to the concentration increase. When ovarian cancer cells were treated with tubulin targeting agents, this phenomenon of S phase increase was frequently observed [19,20]. Meanwhile, 5 μM of paclitaxel and colchicine resulted in drastic cell cycle arrest in the G_2_/M phase (72.6% and 74.3% of cells in this phase, respectively). 

The apoptotic effect of viriditoxin was compared with that of tubulin modulators, such as paclitaxel and colchicine, by flow cytometry (Figure 6). Annexin V FITC-conjugated staining was performed to evaluate the induction of apoptosis in SK-OV-3 cells after treatment with viriditoxin (5, 10, 20, and 40 μM). A notable increase in early apoptotic cell death and a slight increase in late apoptotic cell death were detected in a concentration-dependent manner (Table 3). Paclitaxel and colchicine significantly induced early and late apoptosis.

### 2.5. Viriditoxin Inhibited Migration and Proliferation of SK-OV-3 Cells

The transmigration of cancer cells through the vasculature and their metastasis are attributed to the migratory characteristics of cancer cells. Coordination of dynamic reorganization between the actin filaments and microtubules is essential for cell migration. As one of the components of the cytoskeleton, microtubules are required for maintaining the physical and plastic properties of migrating cells and for both directional and random movements [21]. Tumor metastasis stems from cancer cell migration and colony formation (clonogenicity) [21,22]. Therefore, we analyzed the effects of the microtubule inhibitor viriditoxin on the migratory and clonogenic properties of SK-OV-3 cells. The effect of viriditoxin on cell migration was studied using the scratch wound assay [22]. As a microtubule-modulating agent, viriditoxin considerably inhibited the migration of SK-OV-3 cells at sub-cytotoxic concentrations (2.5–10 μM) (Figure 7). Dotted lines show the boundaries of cells narrowing down after the scratch was made (Figure 7A). The gap between the two dotted lines indicates wound closure distance through cell migration. It was clearly observed that the gap remains unchanged after 24 h owing to the inhibition of cell migration by viriditoxin. Concentration–response measurements further revealed that viriditoxin inhibited cell migration at low concentrations (Figure 7B).

The effect of viriditoxin on colony formation of SK-OV-3 cells was examined using the clonogenic assay (Figure 7C,D). The SK-OV-3 cell colonies were visualized as blue discs by crystal violet staining. It was clearly observed that viriditoxin treatment significantly reduced the colony formation ability of these cells in a concentration-dependent manner. In addition, viriditoxin almost completely inhibited the colony formation at sub-cytotoxic concentration of 2.5 μM. These results revealed that viriditoxin is indeed a microtubule-interfering agent that stabilizes microtubules, thus reducing cell migration and colony formation. 

### 2.6. Binding of Viriditoxin to β-Tubulin

Microtubule binding assay was applied to further confirm the viriditoxin–tubulin interaction as suggested by tubulin polymerization assay (Section 2.2), and to figure out the binding site of viriditoxin to tubulin in comparison with paclitaxel binding site. Viriditoxin (10 μΜ) and excess paclitaxel (20 μΜ) were incubated with tubulin (10 μΜ) at 37 °C for 1 h. As a reference, either viriditoxin (10 μΜ) or paclitaxel (20 μΜ) was incubated with tubulin under the same condition. Tubulins activated by viriditoxin-binding were polymerized into microtubules, and microtubule fraction was separated by centrifugation. Viriditoxin was predominantly detected in the microtubule pellet with the ratio of 99.95% (Table 4). Excess of paclitaxel (20 μΜ) showed 61.44% of tubulin binding, which is to a similar extent to that of viriditoxin (10 μΜ). When viriditoxin (10 μΜ) was incubated with tubulin (10 μΜ) in the presence of excess paclitaxel (20 μΜ), the tubulin-binding of viriditoxin was decreased to 91.97%. Tubulin (10 μΜ) binding of paclitaxel (20 μΜ) was also decreased from 61.44% to 58.99%, in the presence of viriditoxin (10 μΜ). These results indicated that viriditoxin may partially compete with paclitaxel for binding to tubulin.

The interaction between viriditoxin and β-tubulin was further investigated in silico to understand the binding profile of viriditoxin to β-tubulin. Using viriditoxin, paclitaxel, and colchicine as ligands, the binding was computationally examined in the whole β-tubulin protein (protein data base: 1JFF and 1SA0). The crystal structure 1JFF is a tubulin structure with paclitaxel as its ligand, and 1SA0 is a tubulin crystal structure with colchicine as its ligand. According to a docking simulation, viriditoxin was suggested to bind to the site nearby paclitaxel-binding domain (Figure 8). The binding site of colchicine is apparently different from that of paclitaxel and located in the deep pocket, almost inside tubulin (Figure 8) [23]. The distinct antitubulin mechanisms of colchicine and paclitaxel may result from their different tubulin-binding profiles. 

The hypothetical binding mode of viriditoxin was compared with that of paclitaxel (Figure 9). Paclitaxel binds to the pocket of β-tubulin located on the luminal side of the microtubules. This site is composed of the helix H7, strand S7, and the M loop (loops H6-H7, S7-H9) [24]. Viriditoxin was speculated to bind to the paclitaxel-binding site; however, it exhibited hydrogen bonding with Arg^278^, Arg^284^, Gly^370^, and Thr^276^, whereas paclitaxel displayed hydrogen bonding interactions with Gly^370^ and Thr^276^ (Figure 9) [25]. The hydrogen bond (H-bond) with Thr^276^, which is located on the M loop, was predicted for viriditoxin binding [26]. The M loop is a small loop, and it is crucial for promoting tubulin assembly. During tubulin assembly, the M loop is the junction center of current tubulin binding to another tubulin [23]. When the cell is in the G_2_/M phase, microtubules are formed by the polymerization of α- and β-tubulins. In this process, the M loop locks the tubulins one by one to transform them into a polymeric microtubule. The reverse process, depolymerization, is dependent on the M loop as well [23,27]. 

Viriditoxin exhibited van der Waals and hydrophobic interactions with amino acid residues located near the catalytic site. Thus, the binding site and mode of viriditoxin was shown to be similar but subtly distinct from those of paclitaxel.

Meanwhile, the binding site of colchicine is located deep, in the middle of β-tubulin. It is located near both α- and β-tubulins, shaped by helices H7, H8, loop T7, strands S8, and S9 of β-tubulin, also completed by loop T5 of α-tubulin [28]. During microtubule assembly, tubulin dimers undergo a conformational change; they change from a “curved” conformation of the free stage to a “straight” conformation. When colchicine binds to β-tubulin, the T7 loop is switched to release a free space for colchicine binding. The switching of T7 loop hinders the “curved to straight” process, thus inhibiting tubulin assembly and tubulin polymerization during cell mitosis. Docking stimulation results suggested that viriditoxin binds to the paclitaxel-binding domain rather than the colchicine-binding domain. Furthermore, the binding site of viriditoxin appears to overlap rather than be identical with that of paclitaxel. In a co-treatment experiment of viriditoxin and paclitaxel (Table 4), it may be possible that one compound latches on to part of the binding site and partially displace the other, so the binding might be quite complex and binding inhibition by each other would be marginal as shown in Table 4. Possibly, a conformational change caused by binding of one drug would influence the binding of another drug.

In previous studies, viriditoxin was reported to exhibit cytotoxicity against a broad range of human cancer cell lines, including KB, A549, HCT116, SH-SY5Y, and three human prostate cancer cell lines (LNCaP, DU145, and PC3) [9,10]. Viriditoxin was reported to inhibit these prostate cancer cells by inducing multiple modes of growth arrest and cell death coupled with apoptosis and autophagy [9]. In the present study, viriditoxin was proposed to inhibit the proliferation of SK-OV-3 cells by perturbing the microtubule dynamics and leading to G_2_/M phase cell cycle arrest and, finally, cell apoptosis. Microtubule dynamics is the primary target of antimitotic agents such as paclitaxel and colchicine. Paclitaxel stabilizes microtubule polymers, whereas colchicine blocks microtubule polymerization, both leading to inhibition of mitosis. Here, we suggest that viriditoxin stabilized the microtubule polymers in a manner similar to that by paclitaxel. Docking simulation results suggested that viriditoxin binds to the nearby site as that of paclitaxel and forms common hydrogen bonding with Gly^370^ and Thr^276^ including additional hydrogen bonding with Arg^278^ and Arg ^284^. However, the morphology of the stabilized microtubules is distinct from that induced by paclitaxel. Additional biochemical studies are required to elucidate the detailed antitubulin mechanism of viriditoxin and the subtle difference from that of paclitaxel.

The interruption of microtubule dynamics by viriditoxin could also inhibit cell migration and colony formation because microtubules play essential roles in these processes. The scratch wound assay results showed that viriditoxin considerably inhibited the migration of SK-OV-3 cells. Furthermore, it was observed that viriditoxin significantly reduced the colony formation ability of these cells in a concentration-dependent manner. Viriditoxin sufficiently inhibited cell migration and colony formation at sub-cytotoxic concentrations (2.5–10 μΜ and 2.5 μΜ, respectively). In other words, viriditoxin stabilized microtubules and induced apoptosis at higher concentrations, whereas it inhibited cell migration and colony formation at lower concentrations.

## 3. Materials and Methods

### 3.1. Chemicals and Reagents

The natural product viriditoxin was isolated from the marine-derived fungus *P. variotii,* as previously reported [8]. Briefly, *P. variotii* was cultured in a medium containing glucose (20 g/L), malt extract (20 g/L), peptone (1 g/L), and sea salt (26 g/L) at 30 °C on a shaker incubator (155 rpm) for 21 days, in a total volume of 22 L. The culture medium and mycelia were extracted with ethyl acetate (EtOAc). The EtOAc extract was partitioned into aqueous methanol (MeOH) and n-hexane; a yellow precipitate appeared at the interphase of MeOH and n-hexane layers. The yellow precipitate was filtered and identified as viriditoxin by proton nuclear magnetic resonance (^1^H-NMR; 400 MHz). Paclitaxel and colchicine were purchased from Sigma-Aldrich (St. Louis, MO, USA). Primary antibodies against β-tubulin (sc-58886) and bovine anti-mouse IgG-horseradish peroxidase (HRP) (sc-2371) secondary antibodies were purchased from Santa Cruz Biotechnology (Santa Cruz, CA, USA). Alexa Fluor^®^ 488-conjugated anti-mouse IgG secondary antibodies were purchased from Cell Signaling Technology (Danvers, MA, USA). The Annexin V-fluorescein isothiocyanate (FITC) apoptosis detection kit I was purchased from BD Biosciences (San Diego, CA, USA). All other chemicals were purchased from Sigma-Aldrich.

### 3.2. Cell Lines

The SK-OV-3 (human ovarian cancer) and KB (originally isolated from epidermoid carcinoma of the nasopharynx) cell lines were obtained from Korean Cell Line Bank (KCLB^®^, Seoul, Korea). Cells were cultured at 37 °C in 5% CO_2_ humidified incubator and maintained in RPMI 1640 media (HyClone, Logan, UT, USA) containing 100 mg/mL streptomycin, 2.5 mg/L amphotericin B, and 10% heat-inactivated fetal bovine serum (FBS; Gibco-BRL, NY, USA).

### 3.3. Cell Viability Assay

The water-soluble tetrazolium (WST) assay was performed as previously reported to assess cell viability [29]. Briefly, cells were seeded into a 96-well culture plate and allowed to reach 60% confluency prior to treatment with various concentrations of test compounds for 24 h. Cell viabilities were evaluated using WST reagent (EZ-CytoX; Daeil Lab Service Co., Ltd., Seoul, Korea), 10 µL of which was added to each well, followed by incubation at 37 °C for 1 h. The absorbance was read using the iMark Microplate Absorbance Reader (Bio-Rad Laboratories; Hercules, CA, USA) at a wavelength of 450 nm. Changes in the cell morphology were monitored using an optical microscope (OPTINITY; KI-400, Korea).

### 3.4. In Vitro Tubulin Polymerization Assay

The effect of viriditoxin on tubulin polymerization in vitro was evaluated using the commercial Tubulin Polymerization Assay Kit (Cytoskeleton, Denver, CO, USA), according to the manufacturer’s instructions. Briefly, 4 mg/mL tubulin was suspended in G-PEM buffer (80 mM PIPES, 2 mM MgCl_2_, 0.5 mM EGTA, 1.0 mM GTP, and 5% [*v/v*] glycerol; pH 6.9). Next, it was mixed with test compounds, including 10, 50, and 100 µM viriditoxin, 10 µM colchicine, 10 µM paclitaxel, or vehicle dimethyl sulfoxide (DMSO), and transferred to a pre-warmed 96-well plate at 37 °C. The polymerization of tubulin was measured every 5 s for 1 h at 37 °C using a Multiskan GO plate reader at 340 nm (ThermoFisher Scientific, Waltham, MA, USA) [22]. 

### 3.5. Western Blot Analysis

Intracellular tubulin polymerization was examined using Western blotting. The SK-OV-3 cells were seeded into 10-cm Petri dishes at a density of 5 × 10^6^ cells per well; compounds and vehicle were added to the dishes and incubated at 37 °C for 24 h. Cells were subjected to lysis with 100 mM hypotonic buffer (1 mM MgCl_2_, 2 mM EGTA, 1% [*v/v*] NP-40, 2 mM phenylmethylsulfonyl fluoride, 1 µg/mL aprotinin, 2 µg/mL pepstatin, and 20 mM Tris-HCl [pH 6.8]). To obtain cytosolic and cytoskeletal-associated proteins, the particulate fraction was separated from the soluble cytosolic fraction by centrifugation for 1 h at 13,200 rpm at 37 °C. Next, the pellet was dissolved in an equal volume of radioimmunoprecipitation assay (RIPA) buffer. Proteins were separated by 10% sodium dodecyl sulfate-polyacrylamide gel electrophoresis (SDS-PAGE). The distribution of tubulin was analyzed by immunoblotting using anti-β-tubulin antibodies (Santa Cruz Biotech, Dallas, TX, USA) and secondary HRP-conjugated antibodies (Santa Cruz Biotech, Dallas, TX, USA). Signals were developed using the ChemiDoc™ Touch Imaging System (Bio-Rad Laboratories; Hercules, CA, USA) [5]. 

### 3.6. Immunofluorescence

Cells were grown on a confocal dish and treated with the compounds for 24 h. Next, the cells were fixed in 10% formalin for 15 min, washed thrice with phosphate-buffered saline (PBS), treated with 0.5% (*v/v*) Triton X-100/PBS for 15 min, washed thrice with PBS, and, subsequently, blocked with 10% FBS/PBS at 25 °C for 30 min. Cells were incubated with mouse anti-β-tubulin antibodies (Santa Cruz Biotech, Dallas, TX, USA) at 4 °C for 12 h, washed thrice with PBS, incubated for 30 min at room temperature with anti-mouse Alexa 488 secondary antibodies (Cell Signaling technology, Danvers, MA, USA) as a molecular probe, and washed thrice with PBS. Next, they were incubated with PI (10 µg/mL) and RNase (10 µg/mL) at 37 °C for 1 h. Intracellular microtubules and mitotic spindle formation were observed using FluoView FV10i confocal microscope (Olympus, Tokyo, Japan).

### 3.7. Cell Cycle Analysis

The effects of viriditoxin on the distribution of cells in the cell cycle were evaluated by measuring the cellular DNA content. Cancer cells were seeded in 6-well culture plates at a density of 1 × 10^6^ for 24 h, followed by treatment with viriditoxin, paclitaxel, colchicine, or vehicle for 24 h. Cells were harvested by centrifugation, washed with PBS, and fixed in 70% ethanol (EtOH) at −20 °C overnight, and, subsequently, treated with RNase A (100 μg/mL in PBS) for 30 min at 37 °C. After incubation, the cells were stained with PI (10 μg/mL). Stained cells were analyzed by flow cytometry (BD Accuri™ C6, USA).

### 3.8. Annexin V-FITC Binding Assay

The annexin V-FITC binding assay was performed, according to manufacturer’s instructions, using the Annexin V-FITC apoptosis detection kit I (BD Biosciences, San Jose, CA, USA). The cells were treated with viriditoxin for 24 h. The total number of cells was harvested by trypsinization and washed twice with cold PBS. The pellet was suspended at a density of 1 × 10^5^ cells/mL in 1× binding buffer (100 μL) and incubated with FITC-conjugated annexin V and PI at room temperature in the dark for 15 min. Finally, each sample was added to 1× binding buffer (300 μL), followed by analysis using flow cytometry (BD Accuri™ C6, San Jose, CA, USA).

### 3.9. Cell Migration and Clonogenic Assay

Cell migration was studied by the scratch wound assay. Cells were seeded into 6-well plates. A scratch wound was created using a micropipette tip when the cells reached 90% confluency. The cell migration was expressed in terms of wound closure distance after treatment with the drug for 24 h. The cell culture environment was 5% FBS.

For the clonogenic assay, cells at a density of 100 cells/mL were seeded in 6-well plates for 24 h. Next, the cells were treated with viriditoxin or vehicle (DMSO) and allowed to form colonies for 7 days. Colonies were washed with PBS, and cells attached to the plastic surface were fixed in MeOH for 10 min and stained with 0.5% crystal violet for 15 min. The stained cells were subsequently imaged for counting. These were normalized by dissolving in 95% EtOH; the absorption was read at 595 nm.

### 3.10. Microtubule Binding Assay

Tubulin protein (porcine brain source) was prepared in the general tubulin buffer (80 mM PIPES pH 7.0, 2 mM MgCl_2_, 0.5 mM EGTA) with 1 mM GTP. In order to figure out whether viriditoxin compete with paclitaxel at the same binding domain of tubulin protein, viriditoxin (10 μΜ) and excess paclitaxel (20 μΜ) were incubated with tubulin (10 μΜ) at 37 °C for 1 h. As a reference, either viriditoxin (10 μΜ) or paclitaxel (20 μΜ) was separately incubated with tubulin at 37 °C for 1 h. During incubation, ligand-activated tubulin was polymerized into microtubules which could be collected by centrifugation at 90,000× *g* for 20 min at 30 °C, while the non-binding ligand remained in the supernatant. The pellet was resuspended to 10 mM phosphate buffer (PH 7.0). The pellets and supernatants were extracted respectively with excess volume of dichloromethane for three times. The organic layer was dried in vacuum, dissolved in 20 μL DMSO, and then analyzed by reversed-phase HPLC (YMC ODS column 250 mm × 4.6 mm, i.d. 5 μm; wavelength: 220/275 nm; UV detector) using 60% acetonitrile + 0.1% acetic acid as mobile phase. The distribution of each drug in precipitation (p) and supernatant (s) was calculated by the formula: Distritution in precipitation = A_p_/(A_p_ + A_s_) × 100%; Distritution in supernatant = A_s_/(A_p_ + A_s_) × 100%.

### 3.11. Molecular Docking Study

Docking calculations were performed using AutoDock Vina 1.1.2 software (The Scripps Research Institute, La Jolla, CA, USA). Default settings and the Vina scoring function were applied. For ligand preparation, Chem3D Ultra 8.0 software (Cambridge Soft Corporation, Waltham, MA, USA) was used to convert the two-dimensional (2D) structures of candidates into three-dimensional (3D) structural data. Protein coordinates were downloaded from the Protein Data Bank (tubulin: 1SA0 and 1JFF). The analysis and visual investigation of ligand–protein interactions of the docking poses were performed using PyMol v1.5 (Schrödinger LLC, New York, NY, USA). The conformation with the lowest binding energy was used and assumed to be the best binder. The complex conformation of the lowest binding energy was used for molecular dynamics simulations [30].

### 3.12. Statistical Analysis

The significance of intergroup differences was determined by analysis of variance (ANOVA). All data are presented as mean ± standard deviation (SD) of experiments performed in triplicate and in a parallel manner. Statistical significance was accepted for *p-*values < 0.05.

## 4. Conclusions

To summarize, our data showed that viriditoxin inhibited the proliferation of SK-OV-3 cells. The underlying mechanism of cytotoxicity was proposed as stabilization of microtubules in a manner similar to that of paclitaxel. This interruption of microtubule dynamics may lead to cell cycle arrest at the G_2_/M phase and apoptosis. Molecular docking to β-tubulin suggested that viriditoxin binds to the same binding site as that of paclitaxel with slightly different hydrogen bond interactions with key amino acid residues. Henceforth, the morphology of the stabilized microtubules was distinct from that of paclitaxel. Moreover, viriditoxin was demonstrated to inhibit the migration and colony formation ability of SK-OV-3 cells, possibly by interrupting microtubule dynamics. These findings led us to suggest that viriditoxin is a microtubule-interfering agent and could serve as a potential antimitotic and antimetastatic lead.

## Figures and Tables

**Figure 1 marinedrugs-18-00445-f001:**
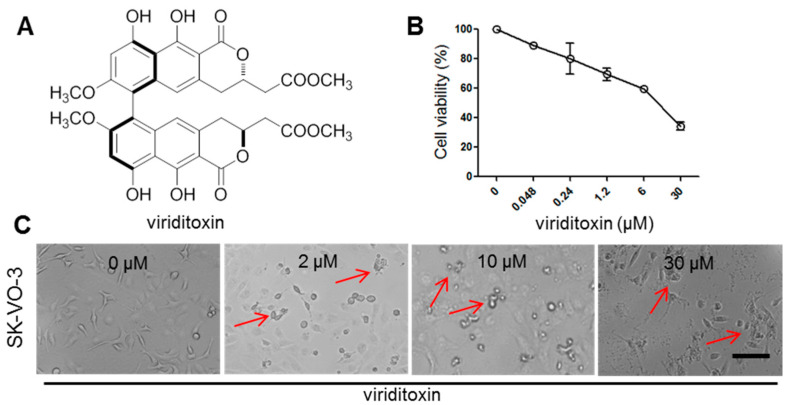
Cytotoxic effects of viriditoxin on SK-OV-3 cancer cells. (**A**) structure of viriditoxin; (**B**) SK-OV-3 cancer cells were treated with viriditoxin at various concentrations (0.048201003330.0 μM) for 24 h; (**C**) morphological changes in SK-OV-3 cancer cells after treatment with viriditoxin for 24 h. Arrows show morphological changes of cells (shrinkage, flattening, rugged periphery of cells). Bar represents a scale of 100 µm.

**Figure 2 marinedrugs-18-00445-f002:**
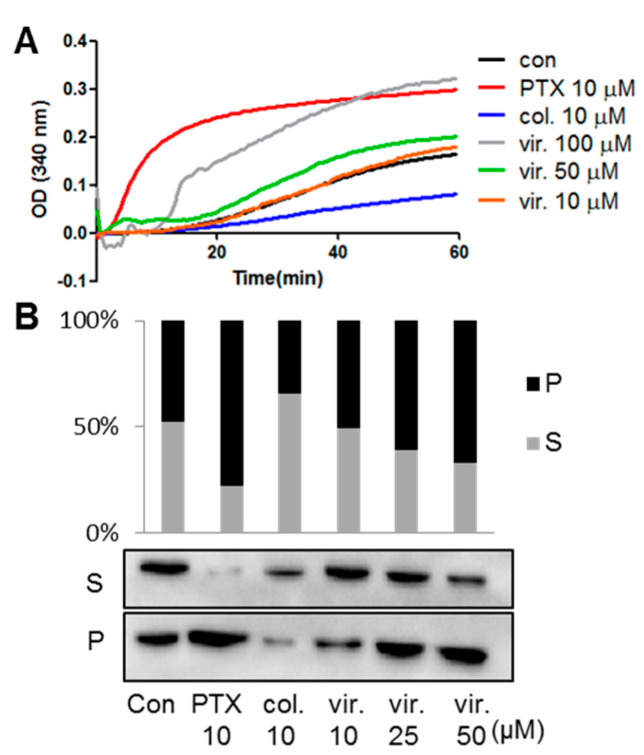
Tubulin polymerization assay. (**A**) Purified porcine brain tubulin was incubated at 37 °C in the presence of viriditoxin (vir.), paclitaxel (PTX), or colchicine (col.). Tubulin polymerization was measured at 340 nm using a microplate spectrophotometer. Polymerized tubulin absorbs UV light at 340 nm. (**B**) Alterations in tubulin polymerization induced by paclitaxel, colchicine, and viriditoxin were measured in the SK-OV-3 cells. The distribution of β-tubulin in soluble (S) and particulate (P) fractions in SK-OV-3 cells was detected by Western blotting after 24 h of drug treatment. Polymerized tubulins are insoluble and particulate in the hypotonic buffer. The results were the means of three times of independent experiments.

**Figure 3 marinedrugs-18-00445-f003:**
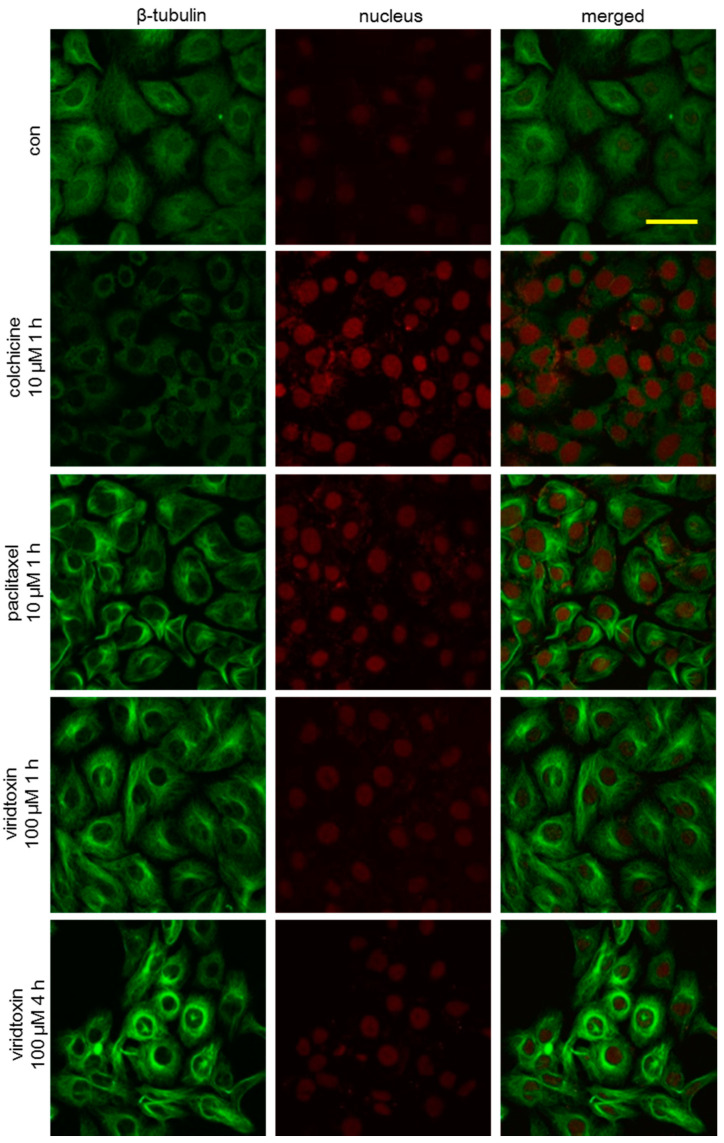
Effect of viriditoxin on the organization of the microtubule cytoskeleton. SK-OV-3 cells were treated with 100 µM viriditoxin, 10 µM paclitaxel, or 10 µM colchicine and monitored after 4 h. Microtubules in the cells were recognized using mouse anti-β-tubulin antibodies, followed by labeling with anti-mouse Alexa 488 secondary antibodies, which were visualized as green fluorescence. Cell nuclei were observed as red fluorescence by propidium iodide (PI) staining. Bar represents a scale of 50 µm.

**Figure 4 marinedrugs-18-00445-f004:**
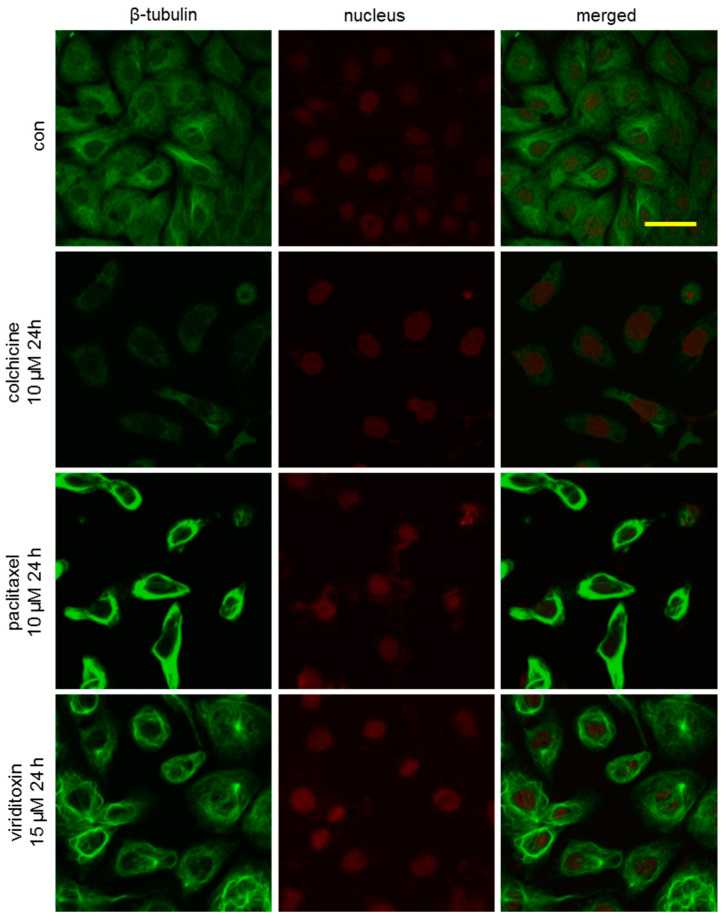
Effect of viriditoxin on the organization of the microtubule cytoskeleton at 24 h. SK-OV-3 cells were treated with viriditoxin 15 µM, paclitaxel 10 µM, or colchicine 10 µM. Bar represents a scale of 50 µm.

**Figure 5 marinedrugs-18-00445-f005:**
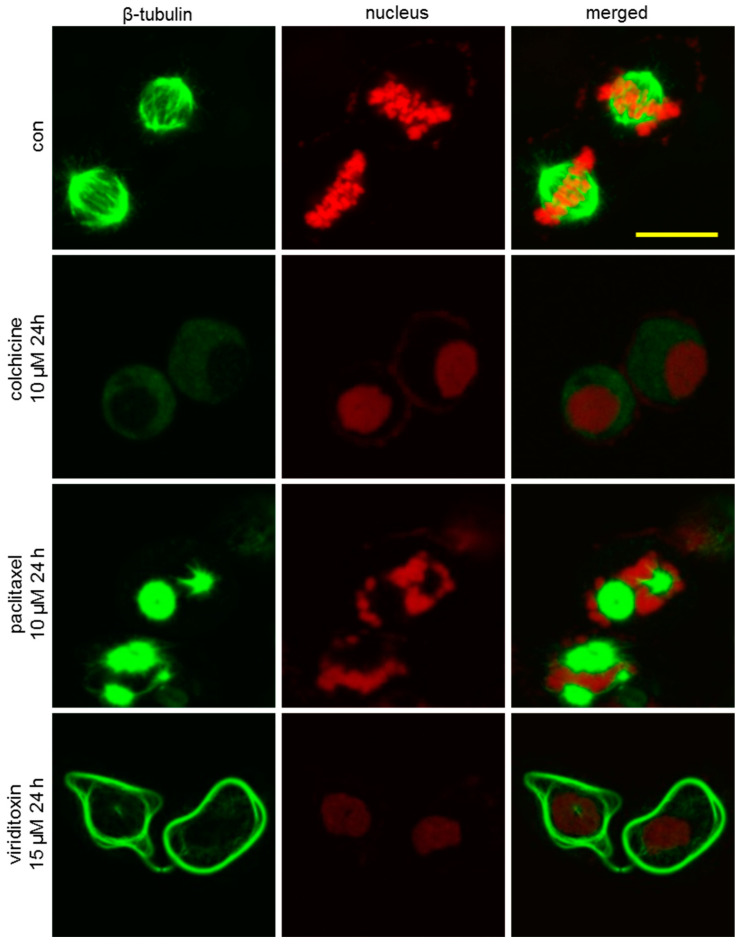
Effect of viriditoxin, paclitaxel, and colchicine on spindle formation in SK-OV-3 cells. Viriditoxin induced abnormal tubulin polymerization with highly dense microtubule linings close to the cell membrane. These pericytoplasmic microtubule structures were apparently distinct from multipolar spindles observed in paclitaxel-treated cells. Bar represents a scale of 25 µm.

**Figure 6 marinedrugs-18-00445-f006:**
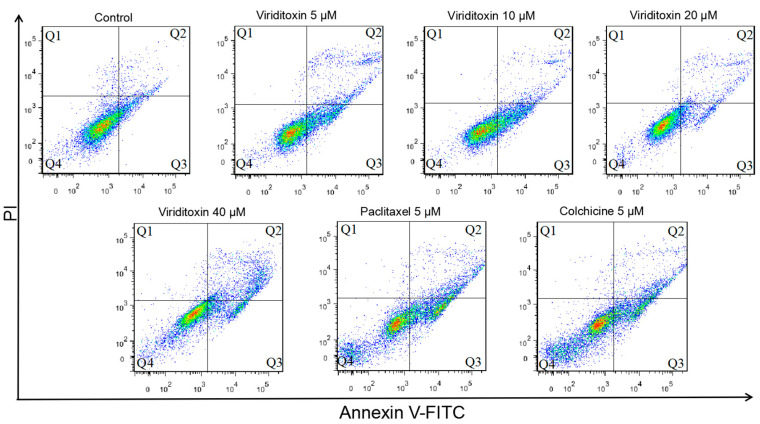
Flow cytometric analysis of apoptosis in SK-OV-3 cells treated with viriditoxin (5–40 μM), paclitaxel (5 μM), or colchicine (5 μM) for 24 h. Cells were stained with annexin V/PI. The Q1 quadrant indicates the population of cells that underwent necrosis, the Q2 quadrant indicates cells in late apoptosis, the Q3 quadrant indicates cells in early apoptosis, and the Q4 quadrant indicates live cells.

**Figure 7 marinedrugs-18-00445-f007:**
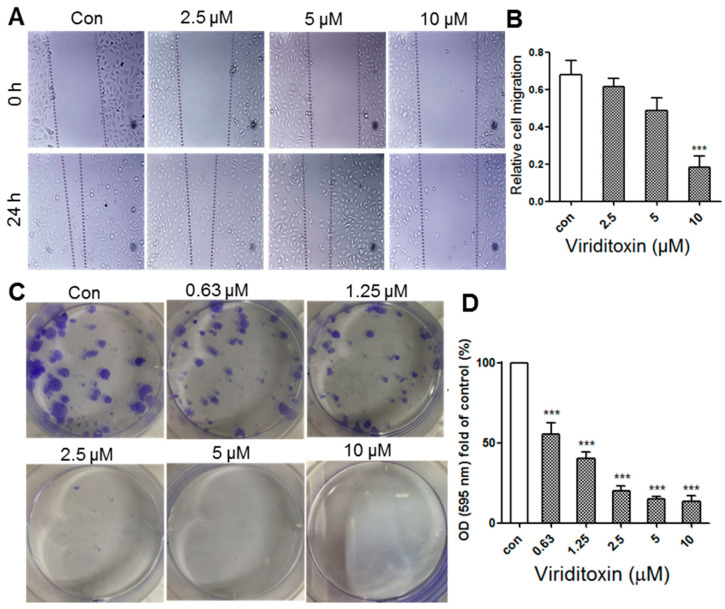
Effect of viriditoxin on cell migration and the colony formation ability of SK-OV-3 cells. (**A**) the effect of viriditoxin on cell migration as demonstrated by the scratch wound assay for 24 h. Dotted lines represent the cell scraping gap. The distance between the two dotted lines indicates the wound closure distance after cell migration upon viriditoxin treatment at 0 h and 24 h; (**B**) histogram presentation of the SK-OV-3 cell migration; (**C**) the effect of viriditoxin on SK-OV-3 cell colony formation. Clonogenicity was evaluated by the monolayer colony formation assay. Representative images show the blue colonies of SK-OV-3 cells stained with crystal violet; (**D**) histogram presentation of the SK-OV-3 cell colony quantification. *** *p* < 0.001 vs. the control group.

**Figure 8 marinedrugs-18-00445-f008:**
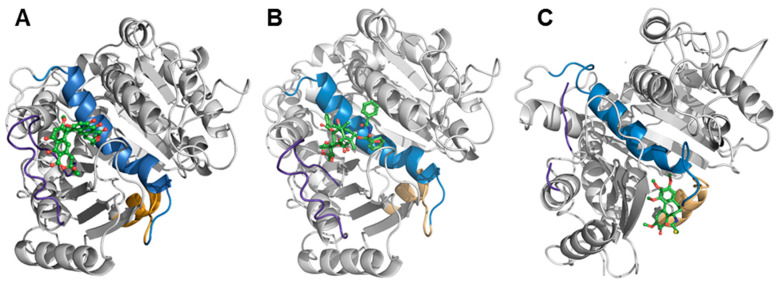
Docking simulation of viriditoxin (**A**), paclitaxel (**B**), and colchicine (**C**) to β-tubulin. The helix 7 (H7) is marked in light blue, the T7 loop is marked in orange, and the M loop (M) is marked in purple. Ligands are shown in green. The binding affinity of colchicine was −7.8 kcal/mol. The affinity scores of paclitaxel and viriditoxin were −8.4 kcal/mol and −8.2 kcal/mol, respectively.

**Figure 9 marinedrugs-18-00445-f009:**
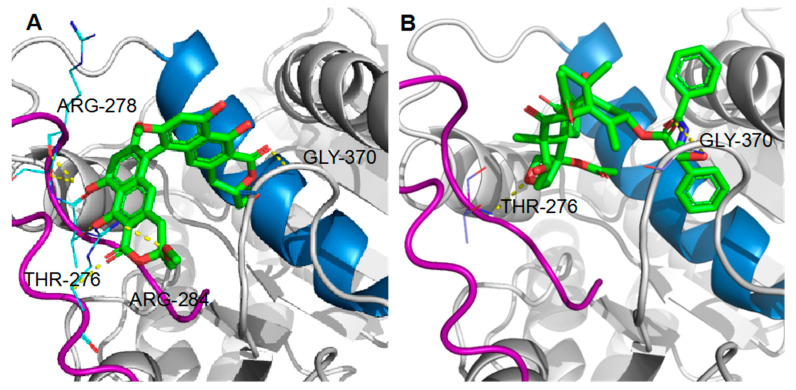
Docking structures of ligand/β-tubulin binding. (**A**) zoomed view of hydrogen bond interactions (yellow dotted lines) between viriditoxin and Gly^370^ and Thr^276^ of β-tubulin; (**B**) zoomed view of hydrogen bond interactions between paclitaxel and Gly^370^ and Thr^276^ of β-tubulin. The helix 7 (H7) is marked in light blue, and the M loop (M) is marked in purple. Ligands are shown in green.

**Table 1 marinedrugs-18-00445-t001:** The cytotoxicity of viriditoxin against human cancer cells at 24 h.

	IC_50_ (µM)	
Cell	Viriditoxin	Paclitaxel
SK-OV-3	14.3	0.24
A549 [10]	5.1	1.9
KB [10]	2.3	
HCT116 [10]	18.0	
SH-SY5Y [10]	12.0	
LNCaP [9]	14.8	
DU145 [9]	18.4	
PC3 [9]	18.7	

SK-OV-3: human ovarian cancer; A549: human lung cancer; KB: human nasopharyngeal cancer; HCT116: human colon cancer; SH-SY5Y: human neuroblastoma; LNCaP, DU145, and PC3: human prostate cancer lines. The cytotoxicity data referenced are cited from previous publications. The IC_50_ values of paclitaxel are rather higher than those described in literature. We used water-soluble tetrazolium (WST) to avoid large deviation of MTT assay. In our experimental condition, the IC_50_ value of paclitaxel was very reproducible and in the μM range.

**Table 2 marinedrugs-18-00445-t002:** Cell cycle distribution (%) in SK-OV-3 cells treated with viriditoxin, paclitaxel, or colchicine for 24 h.

		G_0_/G_1_	S	G_2_/M
Control		63.7 ± 5.0	10.6 ± 2.0	19.3 ± 0.7
Viriditoxin	5 μM	70.7 ± 1.1	7.6 ± 0.9	17.6 ± 0.6
	10 μM	65.7 ± 2.8	8.7 ± 0.7	19.8 ± 0.7
	20 μM	59.7 ± 7.7	11.1 ± 1.4	22.7 ± 4.0
	40 μM	52.0 ± 7.7	12.0 ± 1.6	27.2 ± 3.5
paclitaxel (5 μM)		5.7 ± 1.4	7.4 ± 3.1	72.7 ± 3.7
colchicine (5 μM)		3.1 ± 0.8	7.7 ± 4.8	74.3 ± 8.8

**Table 3 marinedrugs-18-00445-t003:** Distribution of apoptotic populations in SK-OV-3 cells treated with viriditoxin and tubulin modulators.

		Live	Early Apoptosis	Late Apoptosis	Necrosis
Control		90.1 ± 2.6	6.6 ± 3.3	1.6 ± 0.7	1.7 ± 1.5
Viriditoxin	5 μM	82.9 ± 6.2	14.6 ± 6.8	4.2 ± 1.7	0.8 ± 0.4
	10 μM	80.2 ± 7.0	14.8 ± 7.1	3.7 ± 0.7	1.3 ± 1.1
	20 μM	84.3 ± 2.9	8.8 ± 1.6	5.0 ± 1.0	1.9 ± 1.3
	40 μM	76.0 ± 3.9	10.9 ± 0.9	11.0 ± 4.1	2.1 ± 1.2
paclitaxel (5 μM)		73.8 ± 6.6	16.9 ± 4.5	8.0 ± 2.5	1.3 ± 0.5
colchicine (5 μM)		77.4 ± 5.6	15.1 ± 3.6	6.4 ± 2.4	1.0 ± 0.4

SK-OV-3 cells were treated with test tubulin modulators for 24 h and stained with annexin V or PI.

**Table 4 marinedrugs-18-00445-t004:** Competitive binding of viriditoxin and paclitaxel to the tubulin protein.

Distribution (%)	Incubated with a Single Ligand		Incubated with Two Ligands	
	Viriditoxin ^a^	Paclitaxel ^b^	Viriditoxin ^a^	Paclitaxel ^b^
microtubule pellet	99.95 ± 0.07	61.44 ± 13.98	91.97 ± 13.83	58.99 ± 12.31
supernatant	0 ± 0.07	38.56 ± 13.98	8.03 ± 13.83	41.00 ± 12.31

^a^ 10 μM; ^b^ 20μM.

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
