# Peer review of "Viriditoxin Stabilizes Microtubule Polymers in SK-OV-3 Cells and Exhibits Antimitotic and Antimetastatic Potential"

_marinedrugs, 2020, doi:10.3390/md18090445_

Round 1
Reviewer 1 Report
The revised manuscript by Su M et al, has been improved especially in terms of understanding the drug binding site and competition with paclitaxel. Proof reading and correction for English misspelling is necessary before this manuscript is accepted for publication.
Suggestions:
l.227 and l.399: Correct "polymerazed" with "polymerized"
l.394: Correct "procine" with "porcine"
l.400: Change "which could be collect by centrifuge" by "which could be collected by centrifugation"
Author Response
Comment 1: l.227 and l.399: Correct "polymerazed" with "polymerized"
- Answer: Thank you for your comment. The misspelled word "polymerazed" was corrected to "polymerized".
Comment 2: l.394: Correct "procine" with "porcine"
- Answer: Thank you for your comment. The misspelled word " procine " was corrected to " porcine ".
Comment 3: l.400: Change "which could be collect by centrifuge" by "which could be collected by centrifugation"
- Answer: Thank you for your comment. The sentence was correct as suggested.
Reviewer 2 Report
The authors have studied a very interesting drug , viriditoxin, which is known to inhibit the polymerization of FtsZ, which is a prokaryotic protein related to tubulin, which is a eukaryotic protein. They find that viriditoxin enhances tubulin polymerization in vitro and in vivo, as does paclitaxel. So, one very interesting finding is that viriditoxin binds to tubulin and has an opposite effect to what it has with FtsZ. However, both drugs inhibit microtubule dynamic behavior. Another interesting finding that they have made is that the morphologies of the microtubule structures made by viriditoxin and paclitaxel are different, viriditoxin forming a microtubule bundle at the edge of the cell and paclitaxel forming multipolar spindles. Viriditoxin is a very intriguing drug that has the potential to teach us much about structure-function correlations in the tubulin molecule and might have clinical benefits in cancer.
The paper is worthy of publication since it is mostly well-done and the topic is of great potential importance. However,some of the findings do not make sense. Also, the authors need to clarify a few points, especially in one of the figures, and perhaps expand on them in order to make their work more comprehensible and interesting to the reader.
- In the abstract, the authors need to name the fungus.
- Line 39.They should give a reference for the statement of at least 10 members of the tubulin superfamily. The reader may want to follow up and see exactly what this means. I suspect that more than a single reference will be required.
- Line 54.The authors state that the fungus is “jellyfish-derived”. What does that mean exactly? What is the name of the jellyfish and what is the role of the fungus in the life of the jellyfish?
- Line 59.FtsZ forms polymers. The authors should describe the polymers in more detail.
- Lines 59-60.The authors state that FtsZ and tubulin have a close structural homology. The authors should give precise references for this and also explain that the homology is more in the 3D structure than in the primary structure.
- Line 80.In Figure 1C it is somewhat difficult to see the morphological differences between the different panels. Perhaps the authors could spell it out and be more specific. Conceivably some arrows might help.
- Figure 2A is pretty much incomprehensible. There are 6 lines on the figure and 6 labels. However, the structures of the lines do not correspond to the structures shown on the side bar. In other words, the size of the individual dashes on the curves in the figure are not the same size as the dashes next to the labels. It would appear from Figure 2A that 100 micromolar viriditoxin does indeed enhance microtubule assembly since that curve and that of the control are pretty clear. Probably the curve with the longer dashes corresponds to paclitaxel, which also enhances assembly. To continue to guess (the reader should not have to guess), it looks like 50 micromolar viriditoxin causes a slight enhancement of assembly, which is also consistent with the authors’ argument, but colchicine seems to have no effect at all while 10 micromolar viriditoxin seems to inhibit assembly, which is the precise opposite of what the authors are arguing. Turning for help to Figure 2B, one also reaches the conclusion that 10 micromolar viriditoxin inhibits assembly. The authors need to straighten out this situation and also, if different concentrations of viriditoxin have opposite effects on microtubule assembly, they should say so and also come up with a model to explain this.
- Lines 230-234.The degrees of inhibition of binding of paclitaxel and viriditoxin, and vice versa, do not appear to be all that large. The authors did this experiment carefully but one might expect to see more inhibition. However, the binding sites of viriditoxin and paclitaxel appear to overlap rather than be identical. It is conceivable that one drug might latch on to part of the binding site and displace the other so the binding could be quite complex. Perhaps the binding should be measured over a series of times as well as concentrations. The authors may want to consider the possibility of conformational effects on their drug binding results.
- The authors could speculate as to why viriditoxin and paclitaxel cause polymerization of tubulin into different morphologies. Could this be due to different degrees of binding to different isotypes of tubulin that would be present in their experimental system? In other words, if we imagine that isotype A forms the mitotic spindle and isotype B forms microtubules near the cell membrane, and if paclitaxel binds best to the former and viriditoxin to the latter, perhaps that could explain the results. Perhaps each drug has different effects on the binding of microtubule-associated proteins to the microtubule and perhaps this is the reason.
- Finally, the English grammar could be improved.
Author Response
Comment 1: In the abstract, the authors need to name the fungus.
- Answer: Thank you for your comment. The name of the fungus is Paecilomyces variotii, which was added to the abstract.
Comment 2: Line 39.They should give a reference for the statement of at least 10 members of the tubulin superfamily. The reader may want to follow up and see exactly what this means. I suspect that more than a single reference will be required.
- Answer: Thank you for your comment. The relevant references [2] were cited at the line 39, which include 2 references.
Comment 3: Line 54.The authors state that the fungus is “jellyfish-derived”. What does that mean exactly? What is the name of the jellyfish and what is the role of the fungus in the life of the jellyfish?
- Answer: Thank you for your comment. The fungus was isolate from jellyfish Nemopilema nomurai. The isolation method is described as follows:
Following a rinse with sterile sea water, the jellyfish tissue was homogenized and then inoculated on malt extract agar (MEA), which was prepared with 75% sea water, containing glucose (20 g/L), malt extract (20 g/L), agar (20 g/L), peptone (1 g/L), and antibiotics (10,000 units/mL penicillin and 10,000 μg/mL streptomycin, 5 mg/L). Fungi growing out of the jellyfish tissue were separated on the same MEA medium until a pure culture was obtained. Twelve pure fungal strains, including P. variotii, were isolated from the jellyfish.
However, the role of the fungus P. variotii in the life of the jellyfish N. nomurai is still unknown.
Comment 4: Line 59.FtsZ forms polymers. The authors should describe the polymers in more detail.
- Answer: Thank you for your comment. We described the polymers in more detail as “FtsZ forms polymers and assembles into a cytoskeletal ring at the future site of bacterial cell division in the presence of GTP, and along with additional proteins, the ring constricts to divide the cell in to”.
Comment 5: Lines 59-60.The authors state that FtsZ and tubulin have a close structural homology. The authors should give precise references for this and also explain that the homology is more in the 3D structure than in the primary structure.
- Answer: Thank you for your comment. An additional sentence “The N-terminal GTP-binding domains of tubulin and FtsZ are virtually identical in structure, as expected from the substantial sequence identity. Sequence identity is absent from the C-terminal domains, but they also have virtually identical structures [11]” was included with an additional reference.
- b) Erickson, H. P. Atomic structures of tubulin and FtsZ. Trends Cell Biol. 1998, 8, 133-137.
Comment 6: Line 80.In Figure 1C it is somewhat difficult to see the morphological differences between the different panels. Perhaps the authors could spell it out and be more specific. Conceivably some arrows might help.
- Answer: Thank you for your comment. We inserted arrows in Figure 1C as suggested, and a sentence “Arrows show morphological changes of cells (shrinkage, flattening, rugged periphery of cells)” was added to the figure legend.
Comment 7: Figure 2A is pretty much incomprehensible. There are 6 lines on the figure and 6 labels. However, the structures of the lines do not correspond to the structures shown on the side bar. In other words, the size of the individual dashes on the curves in the figure are not the same size as the dashes next to the labels. It would appear from Figure 2A that 100 micromolar viriditoxin does indeed enhance microtubule assembly since that curve and that of the control are pretty clear. Probably the curve with the longer dashes corresponds to paclitaxel, which also enhances assembly. To continue to guess (the reader should not have to guess), it looks like 50 micromolar viriditoxin causes a slight enhancement of assembly, which is also consistent with the authors’ argument, but colchicine seems to have no effect at all while 10 micromolar viriditoxin seems to inhibit assembly, which is the precise opposite of what the authors are arguing. Turning for help to Figure 2B, one also reaches the conclusion that 10 micromolar viriditoxin inhibits assembly. The authors need to straighten out this situation and also, if different concentrations of viriditoxin have opposite effects on microtubule assembly, they should say so and also come up with a model to explain this.
- Answer: Thank you for your valuable comment. We modified Figure 2A by labeling different groups with different colors. The effect of viriditoxin to the tubulin polymerization was found to be negligible at the concentration of 10 μM in Figures 2A and B. For tubulin polymerization assay using SK-OV-3 cells, the western blot result of tubulin distribution was not very sensitive. It was hard to accurately detect the microtubules of the samples since microtubules are easy to depolymerize into soluble tubulins during experimental process. And it was hard to fully collect polymers by centrifugation due to the light weight and invisibility of polymers. However, the histogram is the means of triplicate experiments, and we can see that viriditoxin indeed enhance the tubulin polymerization at a dose dependent manner.
Comment 8: Lines 230-234.The degrees of inhibition of binding of paclitaxel and viriditoxin, and vice versa, do not appear to be all that large. The authors did this experiment carefully but one might expect to see more inhibition. However, the binding sites of viriditoxin and paclitaxel appear to overlap rather than be identical. It is conceivable that one drug might latch on to part of the binding site and displace the other so the binding could be quite complex. Perhaps the binding should be measured over a series of times as well as concentrations. The authors may want to consider the possibility of conformational effects on their drug binding results.
- Answer: Thank you for your valuable comment. As the reviewer mentioned, the binding sites of viriditoxin and paclitaxel appear to overlap rather than be identical. In a binding assay, one drug might latch on to part of the binding site and displace the other so the binding could be quite complex. As the reviewer suggested, we revised corresponding part as follows (lines 285-290) “Furthermore, the binding site of viriditoxin appears to overlap rather than be identical with that of paclitaxel. In a co-treatment experiment of viriditoxin and paclitaxel (Table 4), it may be possible that one compound latch on to part of the binding site and partially displace the other, so the binding might be quite complex and binding inhibition by each other would be marginal as shown in Table 4. Possibly a conformational change caused by binding of one drug would influence the binding of another drug.”
Comment 9:The authors could speculate as to why viriditoxin and paclitaxel cause polymerization of tubulin into different morphologies. Could this be due to different degrees of binding to different isotypes of tubulin that would be present in their experimental system? In other words, if we imagine that isotype A forms the mitotic spindle and isotype B forms microtubules near the cell membrane, and if paclitaxel binds best to the former and viriditoxin to the latter, perhaps that could explain the results. Perhaps each drug has different effects on the binding of microtubule-associated proteins to the microtubule and perhaps this is the reason.
- Answer: Thank you very much. Your valuable comments have inspired us. Your hypothesis is very attractive and accordingly we included a new sentence “This could be due to different binding affinities of viriditoxin and paclitaxel to different isotypes of tubulin that would be present in the experimental system. In other words, if we imagine that different isotypes with distinct functions are co-present and one isotype of tubulin forms mitotic spindle while the other isotype forms microtubules near the cell membrane, and if paclitaxel binds best to the former and viriditoxin to the latter, we may observe aforementioned morphological difference. Or perhaps each drug has different effects on the binding of microtubule-associated proteins to the microtubule (lines 148-154).”
Comment 10: Finally, the English grammar could be improved.
- Answer: Thank you for your comment. We tried to improve our English, especially in grammar.
This manuscript is a resubmission of an earlier submission. The following is a list of the peer review reports and author responses from that submission.
Round 1
Reviewer 1 Report
The authors describe the cytotoxic activity of eukaryotic cells of a marine compound (Viriditoxin) and affirm that it is caused by interaction of the compound with the paclitaxel site of tubulin. Although the hypothesis could be of interest if it is solidly confirmed, I am not convinced by the results presented. Frankly, I feel that the cellular morphology modifications described could be unspecific, and the changes in the cytoskeleton pattern due to the changes in the cell morphology and not the other way around. Also the effect in cell migration and the apoptotic effect could be caused by other pathways.
The main reason for this is the fact that I don´t see a significant increase in the cells stopped in the G2/M phase of the cycle in the presence of the compound, all microtubule modulating agents strongly accumulates the cells in the G2/M phase of the cell cycle. In the experiments described accumulation is minimal while cellular data at the same concentrations shows that the cytoskeleton is largely affected. Why the cells do not accumulate in G2/M if microtubules are so heavily affected?.
It is also striking that the determined IC50 values of Paclitaxel against SK-OV-3 and A549 cells are 3 orders of magnitude over those described in the literature (4.1 nM for A549 1.9 nM for SK-OV-3), these values are routinely determined as well in my laboratory and are in the nM; range, how comes that they are described as uM in Table 1.
Liebmann, J., Cook, J., Lipschultz, C. et al. Cytotoxic studies of paclitaxel (Taxol®) in human tumour cell lines. Br J Cancer 68, 1104–1109 (1993).
Judith A. Smith, Hop Ngo, Miranda C. Martin, Judith K. Wolf, An evaluation of cytotoxicity of the taxane and platinum agents combination treatment in a panel of human ovarian carcinoma cell lines, Gynecologic Oncology, Volume 98, Issue 1, 2005 141-145.
Also notice that although the experiment described in figure 2 indicates that the compound has an influence in the turbidity of the tubulin solution under assembly conditions, an extremely large concentration of the compound is required for a very small effect. This is not compatible with the huge effect seen in Figure 5 at 15 uM of the compound. The effect in vitro is almost not visible at 50 uM.
As said before the study could be of interest if the authors present solid biochemical evidence that the compound actually interacts with tubulin at the paclitaxel site. Such evidence could be easily achieved with a competition assay with a commercially available fluorescent taxane (as an example https://www.thermofisher.com/order/catalog/product/P22310#/P22310). This compound would cosediment with microtubules in a centrifugation assay and this cosedimentation would be inhibited by Viriditoxin if the compound binds to the same site. In such case I would be convinced.
Author Response
Please see the attachement.

Reviewer 2 Report
The manuscript by Su M et al, provides mechanistic details about the cytotoxicity of the natural compound viriditoxin. Results presented in this new manuscript are in line with previous findings from the team in terms of cytotoxicity, apoptosis induction and mitosis blockage in cancer cell lines and they bring new insights about viriditoxin effect on the microtubule cytoskeleton. The study is well designed to assess the microtubule-targeting potential of viriditoxin, and most of the conclusions are supported by the reported results. The manuscript is suitable for publication after a few points are taken care of:
l.36 Humans have eight α-tubulin and nine β-tubulin genes, it could be referred to Ludueña, R. F. and Banerjee, A. (2008).
2.3 Care should be brought to the high concentrations of drugs over a short period of time (24h) used in the study. Far from being clinically relevant, they result in strong effects on the microtubule cytoskeleton which might hide the real mechanism of action. For instance no mention is made about the finely tuned microtubule dynamics that authors could have assessed using a tubulin-GFP construct.
2.4 The authors switch from “microtubule-targeting agents” to “mitotoxin” while they refer to the same compounds (namely paclitaxel and colchicine) officially classified by as “
tubulin modulators”.
Finally, as a general comment, references should be inserted at the end of the sentences not at the beginning.
Author Response
Please see the attachement.

Round 2
Reviewer 1 Report
I see no substantial changes in the manuscript to address my concerns, and I am not convinced by the author’s answers.
I see the potential interest of the manuscript and as discussed by the authors it could be possible that the changes in cellular morphology and cell migration can be related to the effect on tubulin polymerization, but they can be as well related to effect on actin polymerization and the facts that the compound does not arrest cells in the G2/M phase of the cycle and the very limited effect on in vitro tubulin polymerization does not help.
Also we do often perform IC50 experiments with A549 and SK-OV-3 cells and have never observed any problem as those described during the MTT assay. In any case I cannot understand a three orders of magnitude difference between their measurements and ours.
Also I am chemist. I have evaluated a large list of compounds targeting tubulin. In a system with a purified protein and a drug there is no reason for a delay of action time. Either it binds or not. So if the effect of the drug is directly exerted through tubulin the response on a purified system has to be immediate (and I see no actual difference between the control curve and the curve for tubulin incubated with viriditoxin for 8 hours). The facts that it is not immediate in cells suggest that either the active drug is a metabolite or the effect is indirect. As a note the half life of tubulin in solution is 6 hours, so tubulin incubated with viriditoxin for 8 hours should have lost activity.
However I can accept publication if the authors soften their claims AND OFFER ME A SOLID EVIDENCE THAT THE COMPOUND INTERACTS WITH TUBULIN.
I can accept that it might be now difficult to get fluorescent paclitaxel in Korea, however I am 100% sure that an HPLC is available in the chemistry departmen. The following experiment would be acceptable for me.
1.- Incubate 20 uM tubulin with 25 uM vidiriditoxin for 1 hour. Centrifugate to collect the microtubules 90,000 g for 20 min at 37C get the job done. Collect the supernatants by pipetting, and resuspend the pellets in 10 mM phosphate buffer (pH 7.0). Extract the pellets and the supernatants three times with an excess volume of dichloromethane, dry in vacuum, and dissolved the rests in 35 ul of methanol. Analyze the samples by HPLC. If the compound stabilize microtubules it should be bound to the pellet.
2.- Incubate 20 uM tubulin with 25 uM vidiriditoxin and 25 uM paclitaxel for 1 hour. Repeat the analysis as above. If the compound bounds to the paclitaxel site it should be found in the supernatant.